# Results of the Big ANN: NeurIPS'23 competition

**Harsha Vardhan Simhadri[1] harshasi@microsoft.com**
Martin Aumüller[2] maau@itu.dk
Dmitry Baranchuk[3] dbaranchuk@yandex-team.ru
Matthijs Douze[4] matthijs@meta.com
Edo Liberty[5] edo@pinecone.io
Amir Ingber[5] ingber@pinecone.io
Frank Liu[6] frank.liu@zilliz.com
George Williams[7] gwilliams@ieee.org
Ben Landrum[10]   Magdalen Dobson Manohar [11]   Mazin Karjikar[10]   Laxman Dhulipala[10]
Meng Chen[8]   Yue Chen[8]   Rui Ma[8]   Kai Zhang[8]   Yuzheng Cai[8]   Jiayang Shi[8]   Yizhuo Chen[8]   Weiguo Zheng[8]
Zihao Wang[9]   Jie Yin[12]   Ben Huang[12]
[1] Microsoft [2] IT University of Copenhagen [3] Yandex [4] Meta AI Research [5] Pinecone [6] Zilliz
[7] Independent researcher [8] Fudan University [9] Shanghai Jiao Tong University
[10] University of Maryland [11] Carnegie Mellon University [12] Baidu

## Abstract

The 2023 Big ANN Challenge, held at NeurIPS'23, aimed at advancing the state-of-the-art in indexing data structures and search algorithms. It focused for practical variants of Approximate Nearest Neighbor (ANN) search that reflect the growing complexity and diversity of workloads. Unlike prior challenges that emphasized scaling up classical ANN search [21], this competition addressed filtered search, out-of-distribution data, sparse and streaming variants of ANNS. Participants developed and submitted innovative solutions that were evaluated on new standard datasets with constrained computational resources. The results showcased significant improvements in search accuracy and efficiency over industry-standard baselines, with notable contributions from both academic and industrial teams. This paper summarizes the competition tracks, datasets, evaluation metrics, and the innovative approaches of the top-performing submissions, providing insights into the current advancements and future directions in the field of ANN search.

## 1   Introduction

Approximate Nearest Neighbor (ANN) search is an important tool in various fields, including computer vision, natural language processing, information retrieval, and retrieval-augmentation. For example, in the context of Large-Language-Models (LLMs), ANN search is used to add knowledge after model training [14] via retrieval-augmented generation. The necessary similarity search operations such as *nearest neighbor queries* are often required on large datasets, often with *billions* of *high-dimensional, real-valued* vectors, and response times in milliseconds are needed for LLMs to use these in multi-turn reasoning. As result, efficient and accurate ANN search algorithms are essential.

As ANN search becomes commonplace, many variants have become critical in practice. For example, database queries use a combination of vector similarity and predicates over attributes. Multi-modal search involves vectors representing different modalities and thus potentially different distributions. New sparse embedding models are being invented for interpretability and to incorporate text search [9]. Indices are continually updated to reflect changing content and database transactions. These complex

39th Conference on Neural Information Processing Systems (NeurIPS 2025) Track on Datasets and Benchmarks.

| Track | Dataset | Datatype | Dim. | Distance | #Vectors | #Queries | Terms |
|-------|---------|----------|------|----------|----------|----------|-------|
| Filtered | YFCC | uint8 | 192 | $\ell_2$ | 10M | 100K | CC BY 4.0 |
| OOD | Yandex T2I | float32 | 200 | IP | 10M | 100K | CC BY 4.0 |
| Sparse | MSMARCO/SPLADE | float32 | $<10^5$ | IP | 8.8M | 7K | CC BY 4.0 |
| Streaming | MS Turing | float32 | 100 | $\ell_2$ | N/A | N/A | link |

**Table 1:** Overview of datasets used for each of the four tracks, their sizes, dimensions, and other properties.

scenarios are the current reality in the industry, and require indices that work well in constrained computational environments.

Our goal was to shine more light on these variants through a competition with new datasets and baselines, and encourage the research community to develop new indexing and search algorithms, and their optimized implementations. To ensure broad participation and accessibility, the scale of the tasks in the competition was chosen to be large enough to be interesting and small enough to experiment on laptops, small workstations, or virtual machines. The datasets were carefully curated to be representative yet manageable in size, and the evaluation was conducted on standardized Azure virtual machines with limited computational power and memory. Small grants for cloud compute credits provided by Pinecone and AWS further encouraged participation. The competition emphasized open-source contributions, promoting transparency and reproducibility.

This paper summarizes the competition, detailing the specific tracks and datasets used (Section 2), the evaluation metrics employed (Section 3), and the notable approaches taken by the participants (Section 4). By highlighting the advancements made during the challenge, we aim to provide valuable insights into the current state of ANN research and identify promising directions for future work.

**Broader Impact.** While the previous NeurIPS'21 competition on billion-scale approximate nearest neighbor search [21] focused on establishing datasets and the experimental methodology for evaluating large-scale ANN search systems, the present paper proposes novel, industry-motivated search tasks and evaluates the state of the art. We establish clear task definitions, suggest datasets and workloads for them, and introduce the experimental framework that defines the methodology. We believe that this competition had positive impact on this research community. By using small datasets and accessible hardware, as well as issuing generous grants for development, the competition ensured that anyone could participate regardless of their own resources. After the competition, people used our proposal in their own research, see for example [3, 15, 26, 17, 25]. Moreover, the detailed description of competition entries led to top-tier publications such as [5].

**Limitations.** Applications of ANN search, such as ranking or recommendation, can be used towards unethical ends. However, this competition focuses on developing faster algorithms for existing problems, and does not meaningfully enhance any existing capacity for unethical behavior. The limitations of this work are inherent to the task of creating a competition with well-defined evaluation metrics: the metrics and tracks cannot capture every nuance of a robust ANN search algorithm. However, the tracks captured diverse scenarios and used the most widely accepted evaluation metrics in the community.

## 2 Tracks and datasets

The competition consisted of four tracks. In each track, the entry must construct an *index* from a *database* of vectors or dense representations of objects, optimized for the variant of queries applicable to the track. Participants could submit separate entries to one or more of the tracks. Each track uses one dataset listed in Table 1, which also summarizes their properties. All the datasets[1] are available for download from public cloud storage accounts without registration. Except in the case of the streaming track, each dataset consists a set of dataset vectors that are supposed to be indexed, and a set of query vectors. The dataset was made public during the development phase of the competition. For the final evaluation, the dataset vectors remained fixed, while a fresh set of query vectors, unseen to participants, was used. Each track was evaluated independently with its own leader board.

---

[1]All data was collected in compliance with the user agreement of a product or service, and in the case of the MSMARCO dataset, with the consent of crowdsourced editors.

**Query**

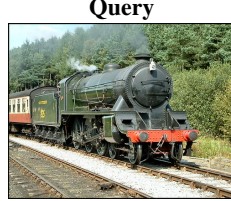

freight
country_GB

**Database**

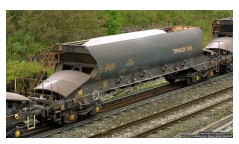

year_2007 month_July
camera_Canon **country_GB** ukrail
tankers horsepower haul britishrail
rail locomotive diesel machine
railway british **freight** work power

camera_Canon **country_GB**
kpa derbyshire transport
rolling rail peak wagon
britain stock railway british
**freight** forest train

**Figure 1:** Example images from the Filtered track, and their associated tags: query (left) and database (right). The images are represented by CLIP embedding vectors.

## 2.1 Filtered Search Track

Searching for entities using a mixture of their semantic properties and associated keywords is natural and pervasive. Examples include searching for a visual match for an image, but from a region or associated with a certain kind of license, or querying articles on arXiv based both on semantic match and time range or author affiliation. This track explored how to build indices that optimize for such queries. The input data comes from the YFCC 100M dataset [24], which consists of embeddings of images from Flickr[2]. We used 10M random images from YFCC100M and embedded them using CLIP embeddings [19]. In addition, we associated to each image a "bag of tags": words extracted from the description, the camera model, the year the picture was taken, and the country. This data was encoded as a sparse vector in the dataset. See Figure 1 for an illustration of datasets and associated tags. The tags are from a vocabulary of 200,386 possible tags. The 100,000 queries consisted of one image embedding and one or two tags. The index returns the images from the database with closest embeddings such that each image's "bag of tags" *must* contain all of the query's tags.

## 2.2 Out-Of-Distribution Track

This track modeled the scenario where the database and query vectors have different distributions in the shared vector space. As observed in [13], existing ANN search indices provide limited recall on such datasets. This track used one such data set – the cross-modal Yandex Text-to-Image 10M. The database is a 10M subset of the Yandex visual search database [3] represented by 200-dimensional image embeddings produced by the Se-ResNext-101 model [10]. The query embeddings corresponded to the user-specified textual search queries and were extracted with a variant of the DSSM model [11].[4]

A simple PCA projection of a sample of query vs. database vectors already shows the discrepancy of distributions. Figure 2 shows the effect of out-of-distribution data. For illustration, let's look at the low-dimensional data, ignoring it's a projection. The left plot shows that many text queries (in the lower-left side of the plot) have the same database nearest neighbor because the database cloud of points does not reach so far to the lower left. This means that the optimal index should be more accurate (or have higher resolution) on part of image database distribution that is most likely to be returned. Similarly, the right plot shows that many database images (in the lower right) will never be returned as the nearest neighbor of a query because they are in an area of the space where there are no queries. This means that an optimal index would just ignore these points altogether. We refer to [13] for characterizations of distribution mismatch for vectors and thus OOD results.

---

[2]Flickr's content policy prohibits offensive images and images that contain identifying information.

[3]The Yandex visual search database removes content where required by law. We were not able to determine whether the dataset creators further restricted identifying or offensive information from the dataset.

[4]This dataset is a point in time and we do not have access to underlying data.

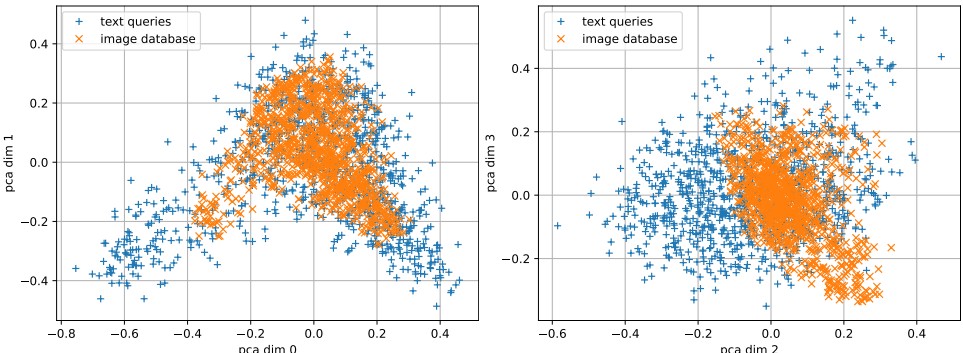

**Figure 2:** PCA projection of 1000 arbitrary query vectors and 1000 database vectors from the OOD dataset. Left: the two first PCA dimensions, right: the two following ones.

### 2.3 Sparse Track

This task was based on the common MSMARCO passage retrieval dataset [18], which has 8,841,823 text passages[5], encoded into sparse vectors using the SPLADE model [9]. The vectors have a large dimension (less than 100,000), but each vector in the base dataset has an average of approximately 120 nonzero elements. The query set was comprised of 6,980 text queries, embedded by the same SPLADE model. The average number of nonzero elements in the query set is approximately 49 (since text queries are generally shorter). Given a sparse query vector, the index should return the top $k$ results according to the maximal inner product between the vectors.

### 2.4 Streaming Track

In this track, the underlying databases evolved over time, and participants were to design an index that supports insertions, deletions and searches. While in practice such indices must support concurrent operations, we allow the index to batch process one class of operations at a time for simplicity. The index starts with zero points and must implement a "runbook" – a sequence of batches of insertion operations, deletion operations, and search commands in a ratio of roughly 4:4:1. This task used a 10 million vector slice of the MS Turing data set released in the previous challenge[6] [21]. In the final run, we used a different runbook than the initial release to avoid participants over-fitting to the runbook. The final runbook consists of 1280 batches of operations consisting of 5 rounds. To generate this, we clustered the 10M points into 64 clusters. Each round consisted of $4 \times 64 = 256$ steps: insert a sample of points from a cluster, search the index using all the queries, delete a fraction of points in the cluster, and search the index again. This simulates distribution drift and point expiration which are both patterns is real workloads. We enforced a memory limit of 8GB to ensure that indices were eliminating the data from the index and a time bound of 1 hour to carry out the whole runbook.

## 3 Evaluation

The entries were run by the organizers on the standard Azure D8lds_v5-series Virtual Machine with 8 vCPUs and 16GB RAM (memory shared by index with OS and standard libraries). Entries for all tracks could use all resources available, except for the streaming track which limited memory to 8GB.

### 3.1 Metrics

Each of the four tasks had an independent leaderboard that participants could submit independent entries to. For each entry, the participants provided a single set of configuration for building an index and a limited list of configurations specifying hyperparameters for querying. The evaluation is carried out with the final query set and the best run is selected. This is akin to the measurements in [6, 1, 21].

---

[5]The passages are anonymized and thus do not contain identifying information, but we were unable to determine whether offensive content was otherwise excluded.

[6]The MS Turing dataset consists of Bing queries and answers. We were not able to determine if it explicitly excludes offensive content and identifying information.

**Search accuracy.** We measured 10-recall@10 where recall is defined as follows:

**Definition 1** *For a query vector $q$ over dataset $P$, suppose that (a) $G \subseteq P$ is the set of actual $k$ nearest neighbors in $P$, and (b) $X \subseteq P$ is the output of a $k'$-ANNS query to an index for $k' \geq k$ nearest neighbors. Then the $k$-recall@$k'$ for the index for query $q$ is $\frac{|X \cap G|}{k}$. Recall for a set of queries refers to the average recall over all queries.*

The definition is easily modified for the streaming scenario and filtered queries. For the streaming scenario, the recall is computed against the set $P$ consisting of all insertions, minus deletions, at the point at which the query was issued to the index. For the filtered search, the recall is computed against the subset of $P$ relevant to the filters specified in the query.

**Throughput.** We measured the overall query throughput on the standardized machine. All queries are provided at once, and the entry could use all the threads available to batch process the queries. We measured the wall clock time between the ingestion of the vectors and when all the results are output. The resulting measure is the number of queries per second (QPS).

**Scoring.** For filtered, out-of-distribution, and sparse tasks, we measured the query throughput of each configuration, and picked the highest throughput that achieved at least 90% 10-recall@10. The leader board lists entries in decreasing throughput at this recall cut-off.

For the streaming scenario, we averaged the recall of queries at various checkpoints over runs that complete in an execution window. That is, the algorithm must complete all insertions, deletions and searches in 1 hour, and only those runs will be scored and ranked by maximum recall across searches.

## 3.2 Evaluation protocol

We extended the benchmarking framework developed by [21] to standardize and automate the evaluation of the four tracks. The framework is open sourced at GitHub[7]. The framework takes care of downloading and preparing the datasets, running the entries, and evaluating the results in terms of providing summarizing metrics and plots. Entries should include code with installation steps to build a Docker container (or provide such a Docker container) and implement the Python API used by the targeted contest track. Each submission was allowed to submit one set of build parameters (per track) and at most 10 sets of hyperparameters defining search-specific behavior. The different hyperparameter settings are intended to strike different speed-accuracy tradeoffs. Except for the streaming track, each submission had to build the index in at most 12 hours using all resources available on the evaluation machine.

The entry submission was handled using Github's pull requests initiated by the authors of an implementation. Authors had the opportunity to give feedback on the experimental runs carried out by the organizers during an interactive round in which organizers reported on the success of the installation and published the result of the evaluation on the public query set. These conversations are recorded in public on the respective pull requests. For the filtered and sparse track, the final evaluation was carried out on a query workload that was kept private to the organizers to avoid overfitted solutions.

**Details of a submission.** A participant has to submit a Python solution[8] that implements a straightforward interface. The evaluation of the sparse, filter, and OOD track contains two parts: In the first part, the evaluation framework provides the dataset $X$ to the implementation. Given $X$, it builds an index $\mathcal{I}$. In the second phase, the evaluation framework presents the query workload $Y$ (in one batch) and asks for the 10 nearest neighbors for each query in $Y$ in $X$ under the task constraints. The implementation will use its search method on $\mathcal{I}$ to produce the resulting set of indices and distances of the approximate solution to the query workload. This set, as well as timing information regarding build and search time, is then stored for further post-processing. For example, in the context of the sparse track, $X$ and $Y$ are CSR matrices to efficiently represent the sparse, high-dimensional vectors. In the context of the filtered track, each vector in $X$ comes with a set of tags, and each vector of $Y$ comes with at most two tags. For the streaming task, there is no preprocessing phase, and the query phase will instead emulate a "runbook" of insert, remove, and search operations, as detailed in the previous section.

---

[7]`https://github.com/harsha-simhadri/big-ann-benchmarks/releases/tag/v0.3.0`

[8]In practice, the performance-critical parts are implemented in a low-level programming language, and a wrapper is used to make this code usable from Python.

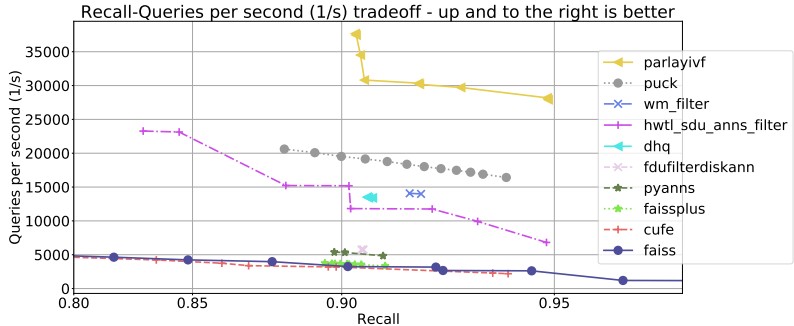

**Figure 3:** Performance of the different algorithms in the filtered track on the private query set.

| Algorithm | parlay | puck | hwtl | wm | dhq | fdu | pyanns | faiss+ | faiss | cufe |
|---|---|---|---|---|---|---|---|---|---|---|
| **QPS (pub)** | 37902 | 19193 | 15059 | 14468 | 13671 | 5680 | 5185 | 3777 | 3033 | 2917 |
| **QPS (priv)** | 37671 | 19153 | 15189 | 14076 | 13517 | 5752 | 5336 | 3625 | 3253 | 2291 |

**Table 2:** Highest QPS achieved by any algorithm in the filtered track with public (pub) and private (priv) query sets, as long as the recall@10 is at least 0.9. Entry names are abbreviated.

## 4 Competition results: baselines and notable approaches

The competition received a total of 26 entries. This section summarizes the competition results for each track, and discusses the techniques used by the track winners and the baselines. The state of the framework and the results, post competition, are captured on Github in v0.3.

### 4.1 Filtered Search Track

The organizers provided the baseline implementation (`faiss`) of the filtered search track, based on Faiss [7]. The baseline operates in two possible modes. In vector-first mode, the search is performed with a Faiss IVF index and vector results that do not satisfy the tag constraint are removed from the result list. In metadata-first mode, the database is reduced to the vectors satisfying the word constraint; in that case the vector search is performed brute force. See [7, Section 6.2] for more details. The baseline is reasonably optimized but uses vanilla Faiss, with parts implemented in Python.

We received ten submissions. Fig. 3 and Table 2 summarize the results of the different algorithms on the Filtered track. The top result is more than 11x faster than the baseline implementation. We observe that there are no major discrepancies between the performance on the public and the private query workload. The participants chose to vary their 10 search hyperparameters to different degrees; all provided usually more than one parameter setting exceeding the target recall.

The winning team ParlayANN (`parlayivf`) used an index whose primary key is the tag associated to each database item. For common tags that are shared by many vectors, a Vamana [23] graph as well as a spatial inverted index are constructed, less common tags are just stored sequentially. At search time, for single-tag queries, the relevant subset of the dataset is accessed immediately and searched, using either a Vamana graph or linear scan. For two-tag queries, three different strategies are used. If one tag corresponds to a set of low cardinality and the other to a set of high cardinality, the smallest tag's elements are intersected with a subset of the largest ones using an efficient bit vector. If both tags correspond to sets of high cardinality, the corresponding spatial indices are used to generate a list of candidates for each tag, and then the intersection of those two candidates is returned. If both tags correspond to sets of low cardinality, the intersection is computed linearly. The queries are ordered to perform similar queries in sequence to improve the cache behavior.

The second-place submission from Baidu (`puck`) is implemented in the Puck library (`https://github.com/baidu/puck`). The index structure has four filtering levels. The first two levels are trained using vector quantization, the last two ones employ product quantization. Each cluster in the levels is labelled with the tags of the vectors in that cluster. This allows to filter out centroids at search time based on the tags.

As visible from the description, the excellent performance of the top participants of this track comes from genuinely handling the filtering constraints with more appropriate data structures. The search for better hyperparameters on the baseline only resulted in minor differences, as visible in the difference between `faiss`, `faissplus`, and `cufe`.

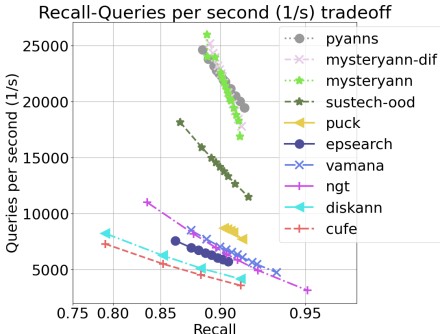

| Algorithm | QPS |
|---|---|
| pyanns | 22296 |
| mysteryann-dif | 22492 |
| sustech-ood | 13772 |
| puck | 8700 |
| vamana | 6753 |
| ngt | 6374 |
| epsearch | 5877 |
| cufe | 3561 |

**Figure 4:** Performance of the different algorithms in the OOD track.

**Table 3:** Highest QPS achieved by any algorithm in the OOD track, as long as the recall@10 is at least 0.9.

## 4.2 Out-Of-Distribution (OOD) Track

The baseline for the OOD track was the in-memory index variant `vamana` in the DiskANN library [20]. While a variant of DiskANN adapted to query distributed exists [13], the baseline does not use those ideas and uses only the points in the database to construct the index.

This track had eight submissions. Fig. 4 and Table 3 show the results of the different algorithms (this track only had a public query set). Due to extremely close performance, MysteryANN (later renamed RoarANN) and PyANNS were declared the joint winners of the track.

RoarANN adopted a graph-based approach, with performance accelerated by scalar quantization and graph reordering. Their graph-based approach took the query vector distribution into account by initially building a bipartite graph between the base distribution and a sample from the query distribution, where each query sample received a directed edge from its top nearest neighbor in the base distribution, and sent $k - 1$ directed edges to its remaining $k$ nearest neighbors in the base distribution. The graph was then projected back into the base distribution. After computing these query-based edges, additional edges were computed using the standard procedure for ANNS graph algorithms in order to form a connected and searchable graph. The approach is published in [5].

PyANNS also used a graph-based approach but did not specifically adapt its algorithm for the out-of-distribution setting. It achieved its winning QPS through careful engineering and optimization of its core library. It used a Vamana graph with a standard greedy search. The search used a scalar quantization of the vectors to 8 bits, with reranking using a 16-bit scalar quantization. The author credits the strong performance of PyANNS to the aforementioned quantization, use of Vector Neural Network Instructions (VNNI), and an adaptive prefetching strategy.

The results show that improvements over the baseline could be achieved in two ways: Through careful algorithm design that adapts the index to the setting of out-of-distribution queries (RoarANN), and through careful implementation engineering (PyANNS).

## 4.3 Sparse Track

The baseline for this track was the `linscan` algorithm [2] available in [12], which is based on an efficient linear scan of an inverted index. Search was accelerated by considering only the largest elements of the query vector, at the expense of accuracy.

We received five submissions each of which used a different technique. Their performance in terms of recall-QPS is shown in Fig.5 and their highest throughput above recall .9 can be found in Table 4.

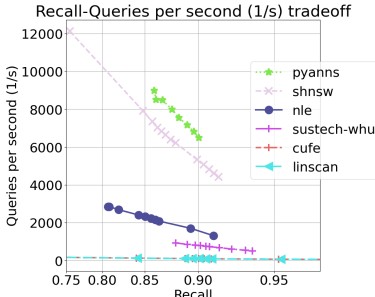

**Figure 5:** Performance of the different algorithms in the sparse track on the private query set.

| Algorithm | QPS (private) | QPS (public) |
|---|---|---|
| pyanns | 6500 | 8732 |
| shnsw | 5078 | 7137 |
| nle | 1313 | 2359 |
| sustech-whu | 788 | 1015 |
| cufe | 98 | 105 |
| linscan | 95 | 93 |

**Table 4:** Highest QPS achieved by any algorithm in the sparse track (private and public query sets), as long as the recall@10 is at least 0.9.

The winning submissions, PyANNS (`pyanns`) and GrassRMA (`shnsw`), both used an HNSW-based graph index [16] but applied different optimizations. PyANNS quantized vector coordinates to 16 bit integers and vector values to 16 bit half-precision floats. Further, during the graph search, the coordinates of vectors in the database were represented as 8-bit integers, and smaller values of the query were pruned away. In order to recover from the accuracy degradation due to the quantization and pruning, the graph search was followed by a refinement step using the full query vector and higher precision base vectors. GrassRMA employed the following optimizations: (1) co-locating coordinates and values of the vector to improve memory access, and (2) keeping an upper and lower bound of the values in the vectors in the index in order to early terminate the dot product calculation. In summary, the winning submissions, won through careful engineering of existing baselines.

We also note the performance differences between the public and private queries. Several algorithms (pyanns, shnsw, sustech-whu) performed around 25% slower on private queries, while nle performed significantly worse (around 45% slower), showing potentially over-fitting on the public queries.

## 4.4 Streaming Search Track

The baseline for this track was the streaming in-memory index variant `diskann` from the DiskANN library[20] using ideas described in [22]. While point insertions are processed eagerly, deletions are processed lazily. A deletion vector is marked as such immediately, but the graph surrounding is not immediately cleaned up. When the index is close to running out of space for inserting new vectors, it runs a "consolidation" method that frees up deleted vectors and re-organizes the graph around deleted nodes to improve search quality. A more detailed analysis of the recall trends of the baseline and HNSW algorithms is provided in the framework.

The streaming track received four entries in total. The entrants were judged by their average recall for queries over the entire runbook, with an hour time limit for executing the runbook, and the official competition results can be found in Table 5.

| Algorithm | Recall |
|---|---|
| puck | 0.985 |
| hwtl_sdu_anns_stream | 0.9674 |
| pyanns | 0.9597 |
| diskann | 0.883 |
| cufe | 0.8189 |

**Table 5:** Recall reported for entries in the official results for the streaming track.

| Algorithm | Recall |
|---|---|
| pyanns | 0.8865 |
| hwtl_sdu_anns_stream | 0.7693 |
| diskann | 0.7218 |
| cufe | 0.6481 |
| puck | 0.0921 |

**Table 6:** Recall of entries after the recall computation was corrected.

The declared winner `puck` by Baidu uses the same baseline implementation as their entry in Filtered search. Insertions were implemented using a natural extension of the build algorithm. Deletions were implemented via an array of flags that allowed deleted points to be filtered during a query.

Unfortunately, more than six months after the competition, we discovered that recall had been calculated incorrectly due to a caching error. The previous results reflected recall at the first snapshot in the runbook rather than averaged over the whole runbook. The error was fixed[9] and the entries

---

[9] `https://github.com/harsha-simhadri/big-ann-benchmarks/pull/280`

were rerun and the recall measured again with the corrected definition[10]. The corrected results are shown in Table 6. The winner under the corrected scoring, PyANNS, used the DiskANN index out-of-the-box with an 8-bit scalar quantization to accelerate the computation which allows more time to search deeper in to the graph index, similar to their entry in the OOD track.

## 5 Discussion

**General remarks.**    Compared to the 2021 issue of the competition, there was more participation and the performance gap between the submissions and the baseline was much wider. We attribute this to (1) the fact that the competition needed accessible hardware which allowed more teams to iterate more often on their algorithms, (2) smaller datasets of 10 million vectors in size, as opposed to billion scale used in the last competition, (3) lesser effort placed in the optimization of the baselines by the organizers, (4) larger interest in this topic given its importance to retrieval-augmented generative AI use cases, and (5) community awareness of the benchmark through citations and prior participation. We interpret this large gap as a sign that there were nontrivial improvements to do on several tracks.

As detailed in the individual discussion, improvements were achieved both through careful algorithmic design choices, for example on how to handle the filtering constraints or how to add information about the "OOD-ness" of the query set to the graph, as well as careful engineering choices on the implementation level, in particular exemplified by the PyANNS submission. No winning entry achieved their performance through hyperparameter tuning of the baseline approach.

There was a considerable difference in the participation level in the individual tasks: In particular the filtered and OOD track received many interesting implementations with a large variety of ideas. On the other hand, the sparse and streaming track received less attention. We speculate that this is due to the difficulty of the tasks and the short timespan in which teams had to come up with a solution.

The filtered search track did restrict the filter predicates to 1 or 2 words. This was done on purpose to narrow down the scope of the competition. However, it also encouraged the participants to develop specialized data structures that may be less interesting for a more general setting. The OOD track encouraged the use of query data samples in the construction of the index as intended.

**Organization glitches.**    Here we identify unforeseen issues in the organization, apart from the technical error in streaming track evaluation, to help future organizers avoid similar pitfalls.

Building a dataset is error prone and sometimes requires making arbitrary choices. Once results on the dataset are published, it is hard to come back on choices made before. We re-used datasets from previous competitions that are frozen, i.e., it is not possible to generate more data from the same distributions. Therefore, it was not possible to get private query sets for all tracks. The process of building the filtered search database was complicated, since it required several stages of metadata extraction, re-balancing, handling of missing data or metadata. In the process we forgot to de-duplicate near exact vectors. This makes the ordering of ground-truth search results arbitrary, and did introduce some jitter in the measurements. However, we could verify that the maximum jitter on recalls is below 0.00015.

Communication with participants required considerable effort – in particular matching registrations received via CMT and pull requests. This made it difficult to reliably identify the affiliation of some (unresponsive) participants. In future iterations, entries are to be submitted with non-anonymous Github accounts and a reference to CMT entries with affiliations.

While there was general agreement on the organizers not competing, there was no written rule published about this, and no exact defininition of an organizer (e.g., would *all* employees of a organizer's company or university be disallowed from competing?). This caused some tensions between organizers and required to take ad-hoc decisions for participants distantly affiliated with organizers. This could have been avoided with clearer rules.

## 6 Conclusion

The Big ANN Challenge at NeurIPS 2023 significantly advanced the field of Approximate Nearest Neighbor (ANN) search by addressing complex real-world scenarios such as filtered, out-of-

---

[10]`https://github.com/harsha-simhadri/big-ann-benchmarks/pull/288`

distribution, sparse and streaming searches. The competition featured significant improvements in search accuracy and efficiency over state-of-the-art baselines through innovative approaches from both academic and industrial participants. Key advancements included improvements in graph-based indexing, quantization techniques, hybrid structures for vector and metadata indexing, and efficient memory access strategies. Advancement have been achieved in two ways: through fundamental algorithmic advances accounting for the task specific setting or by careful engineering and adapting existing implementations. The competition fostered broad participation by emphasizing resource-efficient solutions and open-source contributions.

The Big ANN Challenge has already catalyzed ongoing research efforts in the field, with several new advancements improving the top results of the challenge such as [4], [8], [5] and others. Researchers and practitioners are encouraged to contribute and stay updated with the latest developments through the ongoing leaderboard, accessible at `https://github.com/harsha-simhadri/big-ann-benchmarks/blob/main/neurips23/ongoing_leaderboard/leaderboard.md`.

## 7  Acknowledgements

We are grateful to Dax Pryce for developing the Python wrappers for the diskann library used as a baseline in two tracks, and to Erkang Zhu for a detailed analysis of the recall trends of DiskANN and HNSW under deletions.

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
