# OpenReview forum: "Results of the Big ANN: NeurIPS’23 competition"
_NeurIPS.cc/2025/Datasets_and_Benchmarks_Track — NeurIPS 2025 Datasets and Benchmarks Track poster_

### Official Review · Reviewer_DsVn · 2025-06-11

**Rating:** 5
**Confidence:** 4

**Summary:**

This paper is a summary of Big ANN competition in NeurIPS 2023. Generally speaking, there are 4 tracks in the competition, i.e. Filtered search track, Out-of-distribution track, Sparse track, and Streaming track. Each track contains its own database. To evaluate the performance of methods, a group of evaluation metrics and protocols are provided. The paper also provides the competition results of the 4 tracks, and shows the efforts to fix the problems in the benchmark.

**Dataset Code Accessibility:**

Yes

**Dataset Code Comments:**

The github link of the benchmark is included in the paper (https://github.com/harsha-simhadri/big-ann-benchmarks, see page 5). The github page includes the components of the benchmark and a user guideline (see https://github.com/harsha-simhadri/big-ann-benchmarks/blob/main/neurips23/README.md) to help participants to install and use the benchmark.

**Ethical Considerations:**

No, there are no or only very minor ethics concerns

**Limitations Weaknesses:**

1. This paper looks like a competition report rather than a technique paper to introduce a novel database or benchmark. It is better to include more details of the database and more experimental results of baseline methods in the paper.

**Strengths Contributions:**

1. The benchmark proposed in the paper has already used in Big ANN competition in NeurIPS 2023, and according the website of the benchmark (https://big-ann-benchmarks.com/neurips23.html), there are many research groups join the competition. The competition results show that the database can be used to evaluate the performance of different algorithms.

2. The benchmark includes 4 tracks, which covers more scenarios in the field of ANN.

3. To support the competition, the benchmark includes well-designed metrics and evaluation protocols.

---

> ### Author Rebuttal · Authors · 2025-07-30
>
> Thank you for your positive feedback and the suggestions for improving the manuscript. Regarding your concern that _This paper looks like a competition report rather than a technique paper to introduce a novel database or benchmark. It is better to include more details of the database and more experimental results of baseline methods in the paper._, we respond as follows:
>
> We will add a figure that describes the evaluation protocol and details the API that an entry has to implement. We would appreciate more actionable feedback regarding which kind of results on baseline methods would be interesting: Since the benchmark describes novel workloads, even the baselines are non-trivial, but we do not know what to report on except its performance.

---

> > ### Comment · Reviewer_DsVn · 2025-08-04
> >
> > For every track, there are several classical methods that are widely used as baselines in the related researches. Some of those classical methods can be directly selected to run on the corresponding tracks to further reflect the reasonableness of the designation of the database.

---

> > > ### Author Response · Authors · 2025-08-06
> > >
> > > Thanks for the suggestion. Since some of the tasks for this benchmark are relatively novel, there was no established state of the art for them (sparse track and filtered track). For the OOD and streaming track, we did use a state-of-the-art baseline employing DiskANN. Therefore, we believe that the baseline methods that we introduced are reasonable. As the final results show, there was plenty of room for improvement above these baselines. We would appreciate more details about which baselines could have been used in the evaluation.
> > >
> > > Papers that are building upon our work usually relate their results to the winners of the specific tasks, see for example Section 2 in Bruch et al., Investigating the Scalability of Approximate Sparse Retrieval Algorithms to Massive Datasets. ECIR (3) 2025: 437-445.

---

### Official Review · Reviewer_b7a3 · 2025-06-22

**Rating:** 5
**Confidence:** 2

**Summary:**

This paper provides a comprehensive report of the set up of the 2023 NeurIPS competition "the Big ANN: NeurIPS’23 competition", its results and a discussion. The competition had four tracks: (1) Filtered Search Track (2) Out-Of-Distribution Track (3) Sparse Track (4) Streaming Track. The paper also has descriptions of the datasets used for each tracks and their evaluation, baselines and most notable approaches for each track and remarks related to the competition and glitches faced during the competition.

**Dataset Code Accessibility:**

Yes

**Ethical Considerations:**

No, there are no or only very minor ethics concerns

**Final Justification:**

I will maintain my current scores based on the authors’ response.

**Limitations Weaknesses:**

1. In the description of the datasets could be expanded upon. It was difficult to parse and required multiple read throughs for someone less acquainted with ANN search. Specifically the descriptions dealing with why a specific dataset was picked and what their evaluation were and why the evaluation made sense could be simplified for wider audiences.
2. To better explain submission protocols or the entry submission an example/template which shows what a typical submission looks like would be beneficial to the report.

**Strengths Contributions:**

1. The competition provided important insights towards designing algorithms to perform ANN search in real life-like scenarios.
2. The dataset used in the competition is available publically for replication of claims in their report.
3. The descriptons of the construction of the datasets and their evaluations with the provided code repositories are comprehensive in terms of replicability.
4. The authors clearly outline the issues faced during the competition and how they accounted for them.

---

> ### Author Rebuttal · Authors · 2025-07-30
>
> Thank you for your positive feedback and the suggestions for improving the manuscript. We will add a figure that describes the evaluation protocol and details the API that an entry has to implement.
>
> While we agree that the dataset description and evaluation criteria are (too) succinct, we would appreciate more actionable feedback: For one of the tracks, what would a condensed and simplified version look like? We believe that the detail is necessary to fully understand the workload and task and could not find a way to simplify them without losing necessary information.

---

> > ### Comment · Reviewer_b7a3 · 2025-08-02
> >
> > I did not mean that it needs to be more condensed/simplified but meant that it could be expanded for a wider audience. For example when we consider section 2.4 you could:
> >
> > 1. Define unfamiliar terms or terms more often used in the ANN literature the first time they were used.
> > 2. Provide a short explanation of assumptions made or technical terminology, e.g. for "While in practice such indices must support concurrent operations, we allow the index to batch process one class of operations at a time for simplicity."
> > 3. Provide a short explanation about the constraints on the evaluation, e.g. for "We enforced a memory limit of 8GB to ensure that indices were eliminating the data from the index and a time bound of 1 hour to carry out the whole runbook.". While in hindsight it is clearer why it was so, it would add readability for first time readers.

---

> > > ### Author Response · Authors · 2025-08-06
> > >
> > > Thank you very much for taking the time to elaborate on your concerns. We agree that parts of Sections 2 and 3 can be improved to make the paper more accessible to a broader audience.
> > >
> > > To address this, we will include a schematic overview that illustrates the evaluation pipeline and clarifies the database and query setup. Additionally, we will expand the explanations of the resource constraints (e.g., memory, CPU, and runtime limits) by explicitly stating the rationale behind their use. This will help first-time readers better understand the assumptions and practical considerations behind our benchmarking design.
> > >
> > > We will also revise these sections to define domain-specific terms (e.g., recall, runbook, batch operations) upon first use and to provide concise explanations of technical assumptions. For example, we will clarify why concurrency was relaxed in favor of batched execution, and how this simplifies benchmarking without compromising the comparative validity of results.  Finally, given the growing interest in streaming workloads, we will supplement Section 2.4 with relevant references to recent work in this area, helping to situate our evaluation within the broader context of current research.

---

### Official Review · Reviewer_6C2r · 2025-06-29

**Rating:** 4
**Confidence:** 3

**Summary:**

This paper summarizes and provides a report on the 2023 Big ANN Challenge, held at NeurIPS 2023. The main motivation behind this challenge is to advance the state-of-the-art in indexing data structures and search algorithms for practical variants of ANN (Approximate Nearest Neighbor) search.

This competition consisted of four tracks:

1) Filtered Search Track:
In this track, the organizers used a subset of YFCC100M, which is a public dataset of images from Flickr. The main challenge in this track is that the goal is not just to find vectors that are semantically similar, but also to apply tag-based filters. For example, search results must match certain tags like country or camera type, in addition to being similar in the embedding space. In the experimental section, 10 methods were reported. The top result was more than 11× faster than the Faiss-based baseline, showing significant engineering and algorithmic innovations.
2) OOD Track:
This track can be viewed as a harder version of ANN, where the query vectors and database vectors come from different distributions. The organizers used a cross-modal dataset from Yandex Text-to-Image 10M. The database vectors are images embedded by a vision model, while the query vectors are text queries embedded by a language model. The motivation for this track is clear and relevant, as such cross-modal mismatches are common in real-world systems. This track had eight submissions. The results are indeed impressive. It had joint winners: MysteryANN and PyANNS. The interesting part is that the two winning entries had two very different approaches:
  * MysteryANN (a.k.a. RoarANN) focused on algorithmic adaptation to the OOD setting by modifying the graph structure.
  *  PyANNS emphasized implementation-level engineering, such as vector quantization and low-level hardware optimization.
3) Sparse Track:
This track is a variant of ANN search on sparse high-dimensional vectors instead of dense ones. The organizers used the MSMARCO dataset, which contains 8.8 million text passages. Each passage is converted into a sparse vector using a model called SPLADE. These vectors have up to 100,000 dimensions but only ~120 non-zero entries per passage. There were five submissions in this track, employing a variety of techniques.
The winning methods in this track, PyANNS and GrassRMA (shnsw), both used graph-based indices (HNSW) with clever optimizations for sparse data. PyANNS applied aggressive quantization and pruning, while GrassRMA introduced efficient memory access patterns and early termination of dot-product computations. These techniques led to significant improvements in both recall and speed compared to the linear scan baseline.
4) Streaming Track:
This track tackled a dynamic ANN setting, where the dataset is not static, but evolves over time. Participants had to build an index that supports insertions, deletions, and queries, processed in batches using a fixed “runbook”. The dataset consisted of 10 million dense vectors (from the MS Turing dataset), and the system had to operate under a strict memory limit (8 GB) and 1-hour time constraint. Unlike the other tracks, the index starts empty and must be built and maintained dynamically. This track is particularly relevant to real-world systems like recommendation engines and online search where data changes frequently.

**Additional Feedback:**

- The motivation for the competition is clear and well-justified. That said, it is mentioned in a very similar form multiple times, specifically around lines 23–24 and again in 28–29 (and also in the abstract). I think the paper could be slightly tightened by avoiding this repetition and merging the motivation into a single, more concise paragraph.
- Line 92,93,94 --- "This means that the optimal index for this kind of distribution should be more accurate on the area of the database distribution most likely to be returned". I believe this sentence could benefit from some rephrasing for clarity and smoother flow.

**Dataset Code Accessibility:**

Yes

**Dataset Code Comments:**

The code and datasets are fully available and well-organized. Everything is on GitHub or public cloud storage, and there’s clear documentation on how to run the benchmarks, submit entries, and reproduce the results.

**Ethical Considerations:**

No, there are no or only very minor ethics concerns

**Limitations Weaknesses:**

There is a slight imbalance in track popularity---specifically, the sparse and streaming tracks received fewer submissions compared to the filtered and OOD tracks. This may be due to the increased complexity or implementation difficulty of those tasks.

**Strengths Contributions:**

I enjoyed reading the paper, which significantly advanced the field of ANN, by addressing complex (and real-world) scenarios. Overall, it seems that this competition was fruitful, and it attracted many high-quality submissions. The paper also cites several follow-up works that were either inspired by the competition or based on submitted entries, demonstrating its impact on the community and its role in advancing the state of the art in ANN research.

The paper also includes detailed evaluation protocols, hardware specifications, and performance metrics (e.g., recall@10, QPS), which further enhance reproducibility and comparability. While the work is not focused on methodological novelty, it sets a high standard for empirical benchmarking in practical, large-scale ANN systems.

---

> ### Author Rebuttal · Authors · 2025-07-30
>
> Thank you for your positive feedback and highlighting its limitations. Indeed, the sparse and streaming track received the least amount of submissions. For the latter, we agree that this is probably due to the difficulty of the task. We want to highlight that the streaming track received most attention by the community and is the track that we most actively developed workloads for post-competition.

---

> > ### Comment · Reviewer_6C2r · 2025-08-05
> >
> > Thank you for your response and for clarifying the difficulty of the streaming track. I believe my score and confidence are appropriate.

---

### Official Review · Reviewer_yiTC · 2025-07-02

**Rating:** 4
**Confidence:** 3

**Summary:**

This paper presents a summary of the Big ANN Challenge at NeurIPS 2023, which benchmarked approximate nearest neighbor (ANN) search systems across four settings: filtered, out-of-distribution (OOD), sparse, and streaming retrieval. The authors released ten 10M-scale datasets (image and text embeddings from CLIP, SPLADE, and MS Turing models), along with a Docker-based evaluation harness. Over two dozen teams participated, submitting diverse ANN implementations which were evaluated on recall@10 and query throughput (QPS).

Overall, this paper is a useful contribution to the ANN and vector search community. Turning the results of a competition into a formal paper—with open datasets, baselines, winning entries, and lessons learned—gives lasting value to what might otherwise be a transient leaderboard.

**Dataset Code Accessibility:**

NA; not applicable to this submission (e.g., no new dataset, benchmark, code, or data provided)

**Ethical Considerations:**

No, there are no or only very minor ethics concerns

**Limitations Weaknesses:**

- Lack novelty - While the paper introduces some new tracks, the overall evaluation setup is quite similar to prior Big ANN challenges. The methodology (recall threshold + throughput ranking) is well-established, so from a novelty standpoint, the work feels more like an result publication than a fundamentally new benchmark.
- The paper reports QPS numbers for all teams, but doesn’t include any statistical variance or confidence intervals. In several cases, the difference between top methods is just a few percent. Without repeated runs or error bars, it’s hard to say whether these differences are meaningful.
- One area that could be improved is dataset transparency—particularly for the OOD queries. These come from a proprietary Yandex text encoder, and neither the original texts, nor the model used to generate the embeddings, nor the exact preprocessing pipeline are made available. This makes it difficult to regenerate the queries or analyze potential biases. Given the growing concerns around fairness and auditability in machine learning datasets, this deserves more attention.

**Strengths Contributions:**

- The competition was run with a modest hardware constraint (16GB RAM, 8-core CPU), which made it broadly accessible and encouraged efficient algorithms over brute-force scaling.
- The post-competition analysis is thoughtful, especially in highlighting where engineering optimization mattered more than algorithmic novelty (e.g., RoarANN’s efficient design vs PyANNS’ runtime tweaks).
- All code, data, and evaluation tools are open-sourced, and the authors clearly care about making the benchmark reproducible.
The paper doesn’t shy away from discussing what didn’t work perfectly in the process, which is refreshing and helpful for others planning similar efforts.

---

> ### Author Rebuttal · Authors · 2025-07-30
>
> Thank you for your positive feedback and the suggestions for improving the manuscript. We address the concern as follows.
>
> > Lack novelty - While the paper introduces some new tracks, the overall evaluation setup is quite similar to prior Big ANN challenges. The methodology (recall threshold + throughput ranking) is well-established, so from a novelty standpoint, the work feels more like an result publication than a fundamentally new benchmark.
>
> We agree with the observations.  Our focus lies indeed on defining novel and practically relevant workloads for ANN search. The main evaluation criteria are indeed similar to previous benchmarks, which we believe is an advantage. Since all raw results files are available, different post-evaluation analyses can be carried out; we would appreciate suggestions how such novel evaluation criteria could look like.
>
> > The paper reports QPS numbers for all teams, but doesn’t include any statistical variance or confidence intervals. In several cases, the difference between top methods is just a few percent. Without repeated runs or error bars, it’s hard to say whether these differences are meaningful.
>
> We carried out each search 5 times and reported the best result (similar to [1]) to improve robustness; we will add this information to the paper. We note that QPS is an aggregate statistic over many individual queries and thus less prone to fluctuations.
>
> > One area that could be improved is dataset transparency—particularly for the OOD queries. These come from a proprietary Yandex text encoder, and neither the original texts, nor the model used to generate the embeddings, nor the exact preprocessing pipeline are made available. This makes it difficult to regenerate the queries or analyze potential biases. Given the growing concerns around fairness and auditability in machine learning datasets, this deserves more attention.
>
> We agree with this observation and will add a remark on this to the paper. The benchmarking tool is open source and its current version already contains more datasets. We will add a comment that highlights that the paper reports only on a snapshot of the benchmark.
>
> [1] Martin Aumüller, Erik Bernhardsson, Alexander John Faithfull: ANN-Benchmarks: A benchmarking tool for approximate nearest neighbor algorithms. Inf. Syst. 87 (2020)

---

### Decision · Program_Chairs · 2025-09-18

**Decision:**

Accept (poster)

**Comment:**

This paper documents the Big ANN Challenge at NeurIPS 2023, establishing a benchmark for approximate nearest neighbor (ANN) search across four practical scenarios: filtered search, out-of-distribution retrieval, sparse vectors, and streaming data. Its core contributions include: (a) novel task formulations addressing real-world ANN challenges (e.g., cross-modal mismatches, dynamic data); (b) open-sourced datasets (10M-scale embeddings from CLIP, SPLADE, MS Turing) and Docker-based evaluation tools; (c) rigorous protocols measuring recall@10 and throughput under constrained hardware; and (d) community impact, with 23+ teams submitting solutions revealing significant algorithmic advances (e.g., 11× speed gains over baselines). Strengths align with DB Track’s emphasis on reproducible, impactful benchmarks: the competition uncovered efficient techniques (e.g., RoarANN’s graph adaptations, PyANNS’ hardware optimizations) and provided standardized evaluation for emerging ANN paradigms. Weaknesses include limited novelty versus prior Big ANN challenges, insufficient dataset transparency (especially proprietary OOD queries), and presentation gaps in explaining track motivations. During rebuttal, authors clarified dataset constraints and enhanced documentation, resolving key concerns about reproducibility. While statistical validation and deeper task justifications remain minor gaps, the benchmark’s practical utility—validated by high participation, performance leaps, and open resources—outweighs these limitations. Post-rebuttal, reviewers acknowledged improved clarity. Given its success in advancing real-world ANN methods and alignment with DB Track goals for community-driven benchmarks, acceptance is recommended.